# Hearing Loss in Beta-Thalassemia: Systematic Review

**DOI:** 10.3390/jcm11010102

**Published:** 2021-12-25

**Authors:** Immacolata Tartaglione, Roberta Carfora, Davide Brotto, Maria Rosaria Barillari, Giuseppe Costa, Silverio Perrotta, Renzo Manara

**Affiliations:** 1Department of General and Specialized Surgery for Women and Children, Università degli Studi della Campania “Luigi Vanvitelli”, 80131 Napoli, Italy; immacolata.tartaglione@unicampania.it (I.T.); roberta.carfora@studenti.unicampania.it (R.C.); silverio.perrotta@unicampania.it (S.P.); 2Otorhinolaryngology—Head and Neck Section, Department of Neurosciences, University of Padova, 35122 Padova, Italy; davide.brotto@unipd.it; 3Division of Phoniatrics and Audiology, Department of Mental and Physical Health and Preventive Medicine, University of Campania “Luigi Vanvitelli”, 80138 Naples, Italy; giuseppe.costa@unicampania.it; 4Neuroradiology, Department of Neuroscience, University of Padova, 35128 Padova, Italy; renzo.manara@unipd.it

**Keywords:** hearing loss, thalassemia, iron-chelation

## Abstract

In the last half century, the life expectancy of beta-thalassemia patients has strikingly increased mostly due to regular blood transfusions and chelation treatments. The improved survival, however, has allowed for the emergence of comorbidities, such as hearing loss, with a non-negligible impact on the patients’ quality of life. This thorough review analyzes the acquired knowledge regarding hearing impairment in this hereditary hemoglobinopathy, aiming at defining its prevalence, features, course, and possible disease- or treatment-related pathogenic factors. Following PRISMA criteria, we retrieved 60 studies published between 1979 and 2021. Diagnostic tools and criteria, forms of hearing impairment, correlations with beta-thalassemia phenotypes, age and sex, chelation treatment and laboratory findings including iron overload, were carefully searched, analyzed and summarized. In spite of the relatively high number of studies in the last 40 years, our knowledge is rather limited, and large prospective studies with homogeneous diagnostic tools and criteria are required to define all the aforementioned issues. According to the literature, the overall prevalence rate of hearing impairment is 32.3%; age, sex, and laboratory findings do not seem to correlate with hearing deficits, while the weak relationship with clinical phenotype and chelation treatment seems to highlight the presence of further yet to be identified pathogenic factors.

## 1. Introduction

Beta-thalassemia is a common inherited congenital disorder of hemoglobin production, resulting in hemolytic anemia and multiorgan involvement [1]. Each year, nearly 60,000 beta-thalassemia children are born worldwide, while carriers are estimated to be around 90 million people (1.5% of the global population) [2]. 

In the past centuries, the most severe forms led to early death due to severe anemia. Regular red cell transfusions 60 years ago changed transfusion-dependent β-thalassemia (TDT) from a fatal childhood disorder into a chronic illness [3]. However, while increasing life expectancy, blood transfusions exacerbated multisystem iron overload. In the last decades, the introduction of chelation therapy has further improved life expectancy, allowing for the emergence of late disease- and treatment-related complications with a non-negligible impact on the patients’ quality of life. 

Indeed, the first chelator (deferoxamine) provided evidence of dose-related ototoxicity causing sensorineural hearing loss [4] that led to routine auditory assessments, especially in TDT patients, and to a careful dosage adjustment and/or a reasoned pharmacological shift. Nevertheless, in spite of a more tailored treatment [5], a recent meta-analysis still showed a prevalence of hypoacusia in nearly one third of young transfusion-dependent beta-thalassemia patients (up to 63% in one study) [6]. Even though a large proportion of hearing loss was not sensorineural (up to two thirds [6]) and transient conductive hearing loss is common in the pediatric population, hearing impairment still represents an issue. In fact, hearing loss is known to favor, especially in adulthood, social isolation, depression, and cognitive decline [7], that has repeatedly been described in beta-thalassemia patients [8,9,10,11].

Despite its high prevalence and the several studies available in the literature, the true scenario of hearing loss among beta thalassemia remains rather nebulous. Most studies do not consider incorporating a healthy control group, so that relative prevalence rates are not adjusted with the regional expected prevalence. Besides, findings are strikingly heterogeneous and conflicting with prevalence rates ranging from no hearing problems at all to a hearing deficit in most patients. Discordances also encompass the type of hearing deficit, its severity, and course, with (1) conductive and mixed forms often being neglected, excluded, or not differentiated; (2) variable inclusion criteria from simple high frequency sensorineural dip to severe hypoacusia; (3) scarce information regarding the evolution. In addition, correlation with beta-thalassemia phenotype, chelation treatment, laboratory findings, age, and sex, are poorly defined and the proper management of hearing function varies from yearly investigations to no investigation at all without agreed guidelines about the preferable diagnostic tool.

This comprehensive review aims to make a point about the incidence, etiology, and evolution of this disease- or treatment-related complication highlighting the issues that need further investigations.

## 2. Methods for Literature Review

Following PRISMA guidelines for systematic reviews, the PubMed database, Google Scholar, and www.google.com (accessed on 31 August 2021) were screened up to August 2021, using the following keywords and meshes: “beta-thalassemia” and “hearing” or ‘acoustic‘ or ‘auditory’. All retrieved publications were evaluated, individuating the relevant ones. Duplications of pre-existing data were excluded; only articles in English, German, Spanish, Italian, or French language were included. The reference lists of the selected articles were also screened to identify additional studies. The articles were subdivided among all authors, analyzed, and summarized, excluding non-pertinent papers; all data were thereafter discussed collegially. Appropriate descriptive statistics were applied, when possible, on parametric and non-parametric variables.

## 3. Results

Eighty-five articles were found: 24 were excluded as the full text was not available online nor in the main Italian medical libraries; one was excluded due to Chinese language. Sixty studies were therefore considered; they included 10 case reports and two reviews. The publication years ranged from 1979 to 2021. Seven studies on mixed patients (beta-thalassemia and other anemias) were not considered for our review on the prevalence of hearing loss, as disaggregate data on beta-thalassemia were not retrievable. In the following paragraphs, we present and discuss in detail the literature findings.

### 3.1. Diagnostic Tools and Criteria for Hearing Deficit

In the literature regarding beta-thalassemia, the detection of hearing impairment has been pursued with different, sometimes combined, methods, with likely implications in terms of the observed prevalence rates. Before entering the issue of results, a short overview about diagnostic tools and criteria is necessary.

#### Diagnostic Tools

Clinical interviews widely underestimate hearing loss, as only a minority of beta-thalassemia patients with proven hearing abnormalities complain of hearing impairment [4,12,13,14,15]; in addition, clinical interview is not feasible in young children.

The gold standard to detecting hearing loss in clinical practice is the pure-tone audiometry (PTA). This tool can detect and categorize the degree of hearing loss according to the hearing threshold (mild, moderate, severe, or profound), its type (sensorineural, conductive, and mixed forms), and the audiometric pattern (i.e., flat, sloping, and rising); PTA is even able to detect hearing deficits on a single, generally acute, tone (sensorineural dip).

Even though sensorineural dip does not have a clinical impact, its presence (uni- or bilateral) might represent a warning light of an ongoing pathological process, thus prompting for an audiological follow up to evaluate any progression over time of hearing damage or assess its span on contiguous frequencies. PTA relies on patients’ responses to stimuli and requires cooperative children and adults. Therefore, pure-tone audiometry represents a valid and reliable option for monitoring auditory function in all patients over the age of six. Below this age, and in children who are partially cooperative, the test can be performed in the age-adapted version of behavioral pure-tone audiometry that is feasible though less informative than the pure-tone audiometry.

Otoacoustic emissions, both transiently evoked (TEOAEs) or distortion product (DPOAEs) otoacoustic emissions, are an accurate method for assessing cochlear function, especially the outer hair cell function, and they present several advantages as they are a non-invasive, objective, rapid, easy to use, and sensitive tool. They can be applied even in non-cooperative subjects and seem to be useful for the objective early detection of subclinical hearing loss, even before PTA abnormalities. However, the information given by audiometry is more comprehensive. In fact, DPOAEs are tested at specific intensities (e.g., 35 dB), thus not providing the whole picture of the hearing loss severity. DPOAEs’ absence might be an early sign of hearing loss, but they can also be not detectable for a natural decline of cochlear function. For this reason, DPOAEs should be considered a screening test in non-collaborative patients (e.g., aged less than six) and their findings should always be confirmed with other diagnostic tools (e.g., auditory evoked potentials, PTA). 

Auditory evoked potentials are also objective and can be more informative than otoacoustic emissions, as they could potentially provide a hearing threshold. However, in clinical practice, their use is limited to specific frequencies (2000–4000 Hz) to keep the length of the examination feasible to patients’ compliance. Its use is mostly limited to infants, to non-collaborative patients, or in a research setting.

### 3.2. Diagnostic Criteria in the Beta-Thalassemia Literature

According to the current guidelines, the diagnosis of hearing loss is based on an average hearing threshold increase above 25 dB at 500, 1000, and 2000 Hz (World Health Organization—Grades of Hearing Impairment in different levels of severity). However, ototoxicity primarily affects high frequency domains (e.g., 6000–8000 Hz) and its early detection is improved by considering the decreased performance on single or contiguous high frequencies tested by PTA in one or both ears. For this reason, studies have variably considered hearing impairment above 10 [16], 15 [17], 20 [4,12,13,15,18,19,20,21,22,23,24,25,26,27,28], 25 [14,29,30,31,32,33], 30 [34,35,36], or 40 dB [15] on single, consecutive or multiple frequencies, uni- or bilaterally. In addition, both otoacoustic emissions and auditory evoked potentials might detect objective abnormalities in the auditory pathway that are not strictly related to hearing thresholds. These approaches have further widened the criteria for hearing impairment detection in beta-thalassemia, sometimes without concomitant PTA data [37]. Finally, some studies have reported mixed and conductive deficits or data on single ear, therefore including patients with unilateral hearing abnormalities while other studies have intentionally excluded patients with unilateral hearing impairment or those with conductive or mixed forms [14,25,34]. The lack of uniform diagnostic criteria has obviously greatly conditioned the hearing impairment prevalence in the literature making any comparison among studies very hard and hampering any attempt to define the rate of hearing loss in beta-thalassemia. In addition, most studies lacked a healthy control group that could help understand the role of beta-thalassemia (or its treatments) in the occurrence of hearing deficits.

### 3.3. Beta-Thalassemia Phenotypes and Hearing Loss

Beta-thalassemia has a wide phenotypic spectrum that is classically dichotomized into transfusion-dependent (TDT) and non-transfusion-dependent (NTDT) patients according to hemoglobin levels. This terminology has substituted the previous classification into beta-thalassemia major and intermedia, and the terms are almost interchangeable. However, the two classifications present some minor discrepancies as some studies considered in the “thalassemia intermedia” subgroup also included patients under regular transfusion regimen (e.g., Chen et al. [35]). 

Regarding the hearing loss issue, most studies focused on TDT patients (29 studies, 2766 patients), two of which reported disaggregate data on mixed phenotypes (27 TDT vs. 7 NTDT patients and 51 TDT vs. 20 NTDT) [22,32] and two papers referred only to NTDT (one study with 24 patients that did not report hearing loss prevalence data and a case series including three patients) [38,39]. Several studies had aggregate data regarding mixed thalassemia phenotypes [4,27] or different conditions (e.g., sickle cell disease, Blackfan Diamond anemia, HbE etc.) and the rate of hearing loss was therefore not ascertainable. One TDT study reported data on right and left ear but the number of patients with bilateral or unilateral hearing loss was not retrievable [28].

The mean rate of hearing loss was 29.9 ± 20% among TDT and 31.4 ± 27.9% among NTDT, but across studies both the diagnostic criteria (e.g., pathologic dB threshold, single or average frequency impairment, pure sensorineural or conductive/mixed hearing loss) and the diagnostic tools for hearing loss ascertainment were highly heterogeneous, making the comparison highly arduous. In the three sole studies that included both phenotypes [4,22,32], hearing deficit was less common in NTDT (16/44 vs. 62/108, *p* = 0.02), though the difference was not significant considering only sensorineural hearing deficit (10/44 vs. 36/108, *p* = 0.20).

### 3.4. Types and Prevalence of Auditory Impairment

A recent large review on 1422 Iranian TDT patients [6] found a hearing deficit in 27.3% of TDT patients and reported sensorineural, conductive, and mixed hearing loss in 10.6%, 14.6%, and 9.1%, respectively. Our review confirmed the rate of hearing deficit (1004/3104 patients; 32.3%), although the prevalence rate varied hugely among studies (0–88.2%). 

Most studies focused on sensorineural hearing loss (SNHL), in some cases excluding patients with conductive, mixed, or unilateral SNHL cases, as they were not attributable to chelation therapy-related ototoxicity. On the other hand, some studies included all types of hearing loss without distinguishing among subtypes of hearing loss and without specifying their prevalence rates. The latter studies were excluded from sub-analyses.

SNHL prevalence rate was investigated and reported in 25 studies (mean 24.8 ± 19.0%; range 0–71.3%), while conductive hearing loss (CHL) was reported in 15 studies (mean 10.7 ± 15.9%; range 0–61.8%) and 12 studies reported the mixed form of hearing loss (mean 3.8 ± 5.8%; range 0–14.7%). However, the data are often difficult to be compared as the criteria for hearing deficit varied vastly among studies. In addition, in some studies SNHL showed improvement with chelation therapy adjustment (especially among the oldest studies when the risk of ototoxicity was initially underrecognized) and in other studies CHL was transient or reversible with appropriate treatment. For example, in the study by Albera et al., 22/23 TDT patients with CHL did not present hearing deficits at follow up [20]. The reversibility of CHL is common among children as it is often due to momentary middle ear disease. Extramedullary erythropoiesis involving the ossicular chain has been also repeatedly reported [40,41,42,43].

### 3.5. Age and Hearing

Hearing impairment is commonly associated with aging in the general population, so that a higher prevalence could be expected in adulthood in beta-thalassemia as well. However, other factors such as the disease itself or concomitant therapies and comorbidities might play a pathogenic role in beta-thalassemia, with a possible increased vulnerability in the pediatric age, as well.

Most studies on hearing function in thalassemia were conducted on a mixed population, from pediatric to adult age, and results are presented as a whole group, thus not allowing comparisons between the hearing deficit in children and in adults. Among the 40 studies reporting data on prevalence of HL, 25 described a mixed group (adults and children); to note, these studies recruited young patients, with the mean ages ranging from 9 [23] to 28 years [25]. Five studies did not report the age of studied subjects. 

Three studies enrolled exclusively adults (total 137 subjects, 117/137 TDT) and five only pediatric subjects (total 650 children, all TDT); among the latter, Alzaree et al. did not report data on patients, but on ears [28]. 

The hearing loss prevalence rate was slightly higher in adults (59/137, 43.1%; 52/117 44.4% considering TDT patients) than among children (230/602, 38.2%) though the difference was not significant (*p* = 0.29).

The prevalence ranged in the pediatric population from 15.5% [12] to 73.3% [12,19] in the different studies analyzed, while in adults from 16.6% [31] to 58.8% [32] (see Table 1).

A comparison between adults and children was done with regards to the different hearing loss phenotypes among the TDT patients; an increased prevalence of SNHL was found in adults (40/108, 37.0%, versus 109/602 children, 18.1%, *p* < 0.0001); in contrast, no difference was found in CHL prevalence (8/108 adults, 7.4% and 13/140 children, 9.3%, *p*: 0.59) consistently with a higher rate of middle ear disease among children; mixed HL was noted only in 6/60 adults, not reported in the pediatric studies.

The relationship between age and hearing loss was specifically investigated by some authors, with heterogeneous and inconclusive results; most studies reported no correlation between hearing loss and the age of patients [4,13,14,15,18,24,27,29,30], while some others reported an increased prevalence with age [6,27,28,44], and finally in some studies there was rather a reduction in the prevalence of hearing deficits with increasing age [31,36]. Olivieri et al. and Albera et al. confirmed the results that younger patients had a greater sensorineural hearing loss, and thus they suggested that the ototoxic effect seems to be due to a greater cochlear sensitivity in younger subjects [20,45].

In general, the poor or absent correlation between hearing loss and age seems to suggest that within the beta-thalassemia population, some subjects are more vulnerable, even in the early phase of the disease, while others are almost refractory to auditory impairment.

## 4. Hearing Loss Course

Hearing loss in beta-thalassemia is expected to have a progressive course over time, mostly due to the presumed cumulative toxic effect of chelation therapy. However, very few longitudinal studies are available so far [23,33,34,35,36,57,58] and the scarce data are partly conflicting. Studies investigating hearing loss course suggested that a slow worsening is common [23,33]; however, some authors found no progression during the follow up and one study reported a possible spontaneous improvement during a five-year follow up [36] (Table 2). Chelation therapy discontinuation or dose reduction might reduce the hearing loss in terms of both PTA and subjective impairment [4,23,34,57,58]. However, these observations usually refer to the 1980s and 1990s when chelation therapies often had excessive dosages (up to 120 mg/kg DFO). Nowadays, there is more attention to treatment monitoring and toxic chelation levels are rarely achieved. Nonetheless, the prevalence rate of hearing loss has not dropped. On the contrary, a recent meta-analysis showed an increasing detection of hearing impairment in more recent studies [6]. The high rate of sensorineural hearing loss even in the pediatric population and the scarce increase of prevalence in adulthood (see “age and hearing loss” section) seem to point to an individual cochlear vulnerability so that hearing function is affected early during the disease course. On the contrary, some patients with a history of long-term and high-dose chelation therapy do not present any sign of hearing involvement, suggesting that other yet unidentified environmental and genetic factors might play a role. 

## 5. Iron Chelation and Hearing Loss

The first report of hearing deficit in beta thalassemia came in 1979 from De Virgiliis [18], who reported in a group of 75 TDT children a rate of 73.3% hearing loss, including both SNHL and CHL; all patients were receiving chelation therapy with deferoxamine (DFO), which at that time was still administered intramuscular, with doses ranging from 750 to 1000 mg/day (no dose/kg reported). Despite this, De Virgiliis suggested that iron overload per se was the likely cause of hearing loss, together with the bone marrow expansion. Thereafter, some cases of suspected DFO-related ototoxicity were described [46,47], raising the interest in monitoring hearing function in beta-thalassemia patients undergoing chelation. 

Indeed, Porter et al. found SNHL only in patients treated with iron chelation (DFO) and described a direct correlation with the chelation dosage but also an inverse one with ferritin levels [4]. According to these findings, the authors concluded that a therapeutic index (TI: (dose in mg/kg)/serum ferritin) greater than 0.025 for more than three months was unsafe for hearing function and had to be prevented; this was then suggested to be safe at 0.027 [36]. The correlation with DFO dose was supported by a pharmacokinetic study in TDT children with and without neurotoxicity (hearing/vision loss), showing that the administered dose was higher in “neurotoxic” subjects [59].

The available literature however is not in agreement with regards to iron chelation’s role in causing hearing loss; our analysis shows that a correlation with the dose of the iron chelators is reported mostly when doses are higher than the recommended (i.e., greater than 40 mg/kg/d for DFO). 

Most studies were conducted on patients receiving DFO, so that data are scarce for the other two iron chelators; only in two studies deferasirox was the sole chelator used, and no study was found on patients on deferiprone monotherapy; six studies included patients on different iron chelation drugs, but they did not report separate data for each chelator (Table 1). In addition, it is not clear if patients treated with deferasirox or deferiprone were previously treated with DFO.

Lastly, no study compared the rate of hearing loss among the three different chelators or analyzed the prevalence rate among patients naive for iron chelation. Interestingly, abnormal BAEPs were registered in a group of NTDT subjects, not on iron chelation [38,50], and recently, SNHL was reported even among some NTDT patients who were not on chelation treatment [32], highlighting that SNHL could also be a complication of beta-thalassemia itself. Indeed, beta-thalassemia patients (regardless of disease severity and chelation therapy) showed brain perfusion changes at the level of the primary auditory cortex suggesting a more complex pathogenesis of hearing dysfunction [32].

Further investigations are therefore warranted also in not-chelated and not-transfused beta thalassemia patients to achieve more information about the role of chelation treatment in the SNHL onset. 

## 6. Iron Overload and Hearing Loss

One of the main complications in TDT patients is the secondary hemochromatosis due to tissue iron deposition occurring during the disease course and the prolonged transfusion regimen. Current guidelines recommend that therapy must be started after serum ferritin levels are above 1000 μg/L. The long-term control of serum ferritin has been linked to protection from heart disease and to improved survival if levels are consistently less than 2500 μg/L [60], with even better outcomes at levels <1000 μg/L [61]. The excessive iron deposition in the heart, liver and endocrine glands is responsible for heart failure, liver fibrosis and cirrhosis, diabetes mellitus, hypogonadism, growth failure, sexual immaturity, and immunological alterations. 

Serum ferritin levels have been also implicated with hearing impairment development, but the relationship is still controversial. Most authors found no significant ferritin level differences between patients with and without hearing loss [12,14,17,20,23,24,25,29,30,35,44,48,58,62].

In the study by Porter et al. in 1989, low serum ferritin appeared as a risk factor for SNHL, since all patients with hearing impairment had serum ferritin below 2000 μg/L [4]. This association was confirmed in a few subsequent studies [26,27,28,45] in patients with thalassemia undergoing long-term transfusion therapy, and it was attributed to the invasive treatment of DFO to reduce iron overload in patients with ferritin levels above 3000 μg/L [6], or to the excessive depletion of metals in overtreated patients [49]. However, two small sample studies had opposite results suggesting that iron overload could also be associated with auditory deficiency at high frequencies [18,31].

Thus, according to the available literature, both iron overload and excessive iron-chelating therapy should be prevented, and a tailored chelation treatment should be adopted.

## 7. Anemia and Hearing Loss

The possible relationship between hearing impairment and the severity of anemia (hemoglobin levels or number of transfusions) was investigated in eight papers. More than 1200 patients undergoing regular transfusion regimen were analyzed with the following results: abnormal PTA results and hearing loss were non-related with pre-transfusion hemoglobin [26,44], duration of transfusion therapy [26], age at the first blood transfusion [13], mean hemoglobin in the last three months [24], mean annual hemoglobin values [18,35] or unit of blood received per year [35].

In a small sample of TDT patients, a hearing threshold decrease correlated with hemoglobin levels, transfusions per year, and duration since the last transfusion. These results were explained by a lower formation of insoluble alpha chains tetramers when hemoglobin increases, but no other study supported this hypothesis. Furthermore, as total transfusion numbers increased, hearing thresholds at PTA and high frequency audiometry decreased [31].

## 8. Sex and Hearing Loss

Several studies have investigated the relationship between sex and hearing loss in order to identify additional risk factors for ototoxicity. In nine articles (757 beta-thalassemic patients), the correlation between sex and hearing impairment was analyzed: most of these studies failed to detect a significant correlation [6,17,23,24,26,27,28]. Only two papers showed a possible association between sex and hearing loss and found respectively an increased [14] and a decreased [31] severity in males. Therefore, there seems to be no relationship between sex and hearing loss in beta thalassemia.

## 9. Tinnitus

Tinnitus is one of the most common hearing disorders, with wide-ranging risk factors including age, hearing loss, noise exposure, inflammatory diseases, psychosocial distress, and ototoxic drugs. The latter may induce both reversible and irreversible damage of the inner ear structures causing tinnitus, with or without hearing loss and balance problems [63]. These ototoxic effects seem to be related to the duration of therapy, route of administration, dosage, individual sensitivity, genetic predisposition, and altered renal and hepatic functions [64].

Data on tinnitus in thalassemic patients are rather scanty and often dispersed in the results sections. They were acquired only by means of an anamnestic interview; no study applied specific instrumental examinations or standardized questionnaires. For these reasons tinnitus is often poorly characterized in terms of its severity, disability, correlation with age, or association with hearing loss. In addition, no study included a healthy control group or mentioned the prevalence of tinnitus in the general population. In beta-thalassemia, tinnitus prevalence rate ranged widely from 3.3% to 38% [4,15,30,35,44,46,57,62,65], so that tinnitus was defined as either rare or common, up to being the most common hearing symptom in thalassemic patients, both in subjects with normal audiogram and in those with hearing loss.

As tinnitus is often underreported by patients and it is frequently associated with ototoxicity, even in the absence of hearing loss, dedicated investigations by means of standardized questionnaires or acufenometry (frequency measure; intensity measure and the minimum level of tinnitus masking) are warranted to improve the follow up of beta-thalassemic patients.

## 10. Audiological Monitoring and Management

Audiological testing has become routinary practice in beta-thalassemia for multiple reasons, at least in the transfusion-dependent form. First, the need for potentially ototoxic long-lasting therapies and the ageing due to an improved life expectancy are expected to determine an increasing prevalence of hearing loss in beta-thalassemia patients. Second, hearing loss might be a precious warning light of toxicity of the ongoing therapy, thus allowing for prompt dosage adjustment; this approach was proven to often revert or reduce the perceived hearing loss [23,57]. Third, untreated hearing loss might cause a wide range of consequences, from social isolation to depression, and might predispose to an earlier onset of dementia [66].

Few longitudinal studies are available (see “hearing loss course” section). Currently, guidelines on audiological assessment and management are missing. As during the clinical trials of iron chelation, audiological assessment was done yearly, this approach has been automatically translated in the clinical practice, even though there is no supporting evidence. Notably, even audiological data obtained during trials have been not or very scarcely published so that the reason for such a strict follow up remains obscure [67]. 

Since the outset of the hearing loss is still unpredictable, audiological testing is suggested before the start of treatment to assess the hearing threshold. Audiological evaluation should be considered mandatory in the case of the onset of hearing disorders (subjective hearing loss, fullness, tinnitus). Integrating previous suggestions [33], in the case of patients with normal hearing function, it seems to be reasonable to plan following tests each year in the pediatric age and every two years in adolescence (mostly due to the dramatic impact of undiagnosed hearing loss on scholastic/learning performance). In adulthood, audiological testing seems to be reasonable every 3–5 years even in patients not reporting hearing impairment. The detection of a rapidly progressive hearing loss should lead to a stricter follow up. Hearing aids should be promptly considered whenever hearing loss is impairing daily social life [4].

## 11. Conclusions

Despite the large number of studies addressing the topic of hearing loss in beta-thalassemia, the discrepancies in recruitment and diagnostic criteria do not allow the obtainment of a reliable and precise picture of this problem. Future longitudinal studies with a detailed description of sample, treatment, and hearing deficit will help understand the pathogenesis, the prevalence, and the best management of hearing impairment in beta-thalassemia.

## Figures and Tables

**Table 1 jcm-11-00102-t001:** Literature overview of hearing impairment in beta-thalassemia.

Author and Year	#	Age	Phenotype	ICT	HL	SNHL	CHL	MHL	Method	HL Definition
De Virgiliis et al., 1979 [18]	75	ped.	TDT	DFO	73.3%	57.3%	16%	n.r.	PTA	>20 dB in any frequency
Marsh et al., 1981 [46]	1	adult	NTDT	DFO	0	0	0	0	clinical interview	Not Applicable
Orton et al., 1985 [47]	2	n.r.	TDT	DFO	100%	100%	0	0	n.r.	n.r.
Barrat et al., 1987 [19]	25	mixed	TDT	DFO	36%	36%	0	0	PTA	>20 dB in any frequency
Albera et al., 1988 [20]	153	ped.	TDT	DFO	71%	71%	15%	n.r.	PTA	>20 dB in any frequency
Masala et al., 1988 [16]	100	mixed	TDT	DFO	24%	12%	12%	n.r.	PTA	>10 dB in any frequency
Cohen et al., 1990 [21]	27	mixed	TDT	DFO	0	0	0	0	PTA	>20 dB in any frequency
Porter et al., 1989 [4] *^	30	mixed	TDT	DFO	23.3%	23.3%	6.7%	n.r.	PTA	>20 dB in any frequency
Porter et al., 1989 [4] *^	17	mixed	NTDT	DFO, none	23.5%	11.8%	17.6%	n.r.	PTA	>20 dB in any frequency
Wonke et al., 1989 [34]	50	mixed	TDT	DFO, Ca-DTPA	26%	26%	n.r.	n.r.	PTA	>30 dB in any frequency
Triantafyllou et al., 1990 [48]	120	mixed	TDT	DFO	36.6%	10%	15%	12.2%	n.r.	n.r.
Argiolu et al., 1991 [12]	309	ped.	TDT	DFO	15.5%	15.5%	n.r.	n.r.	PTA	>20 dB in any frequency above 2000 Hz
Cuda et al., 1991 [35]	50	mixed	TDT	DFO	26%	n.r.	n.r.	n.r.	PTA	>30 dB in any frequency or >25 dB in at least 2 frequencies
Sheikha et al., 1992 [39]	3	mixed	NTDT	n.r.	100%	66.6%	33.3%	0	n.r.	n.r.
Wong et al., 1993 [49]	34	mixed	TDT	DFO	11.8%	n.r.	n.r.	n.r.	BAEP	n.r.
Onerci et al., 1994 [22] *	27	mixed	TDT	DFO	92.6%	14.8%	63%	14.8%	PTA	>20 dB in any frequency
Onerci et al., 1994 [22] *	7	mixed	NTDT	none	71.4%	0	57.1%	14.2%	PTA	>20 dB in any frequency
Sacco et al., 1994 [36]	36	mixed	TDT	DFO	19.4	19.4%	n.r.	n.r.	PTA	>30 dB in any frequency
Kontzoglou et al., 1996 [23]	88	mixed	TDT	DFO	27.3%	27.3%	0	0	PTA	>20 dB in any frequency
Levine et al., 1997 [50]	2	adult	TDT	DFO	50%	50%	0	0	n.r.	n.r.
Meara et al., 1998 [41]	1	adult	NTDT	n.r.	100%	0	100%	0	n.r.	n.r.
Ambrosetti et al., 2000 [29]	57	adult	TDT	DFO	33.3%	26.3%	7%	0	PTA	>25 dB in any frequency
Passat et al., 2001 [37]	65	ped.	TDT	nr	29.2%	27.7%	1.5%	0	BAEP	>30 dB
Karimi et al., 2002 [24]	128	n.r.	TDT	DFO	56%	11.7	n.r.	n.r.	PTA	>20 dB in any frequency
Chen et al., 2005 [30]	25	mixed	TDT	DFO	20%	20%	n.r.	n.r.	PTA	>25 dB in any frequency
Sonbolestan et al., 2005 [14]	160	n.r.	TDT	DFO	48.7%	n.r.	n.r.	n.r.	PTA	>25 dB in any frequency or >10 dB in two sequential frequencies
Berjis et al., 2007 [15]	160	mixed	TDT	DFO	50%	50%	n.r.	n.r.	PTA	>20 dB in two consecutive frequencies, or > 40 dB in one frequency
Budak et al., 2008 [31]	9	adult	TDT	DFO	16.6%	n.r.	5.5%	11.1%	PTA	>25 dB in any frequency
Delehaye et al., 2008 [25]	60	mixed	TDT	DFO	50%	50%	0	0	PTA, DPOAE	>20 dB at PTAv or increase of at least 20 dB in any frequency at follow up
Shamsian et al., 2008 [17]	67	mixed	TDT	DFO	7.4%	7.4%	0	0	PTA	>15 dB in any frequency
Thio et al., 2008 [42]	1	ped.	TDT	DFO	100%	0	100%	0	n.r.	n.r.
Faramarzi et al., 2010 [13]	293	mixed	TDT	DFO	3.5%	3.5%	0	0	PTA	>20 dB in any frequency
Vir et al., 2010 [51]	26	mixed	TDT	DFO, DFP	n.r.	n.r.	n.r.	n.r.	PTA	>= 26 dB in any frequency
Chao et al., 2013 [26]	37	mixed	TDT	DFO, DFP	35.1%	n.r.	n.r.	n.r.	PTA	>20 dB in any frequency
Uygun et al., 2013 [52]	169	mixed	TDT	mixed	14%	n.r.	n.r.	n.r.	n.r.	n.r.
Pereira da Silva et al., 2015 [53]	2	ped.	mixed	none	100%	0	100%	0	PTA	n.r.
Osma et al., 2015 [27]	159	mixed	mixed	mixed	39%	39%	n.r.	n.r.	PTA	>20 dB in any frequency
Bhardwaj et al., 2016 [33]	30	n.r.	TDT	mixed	23.3%	n.r.	n.r.	n.r.	PTA, DPOAE	Average of 2, 4 and 8 kHz >25 dB
Badfar et al., 2017 [6]	1422	n.r.	TDT	DFO	27.3%	10.6%	14.6%	9.1%	n.r.	n.r.
Lanigan et al., 2017 [40]	1	ped.	TDT	DFX + DFO	100%	0	100%	0	PTA, OAE	n.r.
Hasan et al., 2018 [54]	23	mixed	TDT	DFO, none	26.1%	4.3%	21,7%	n.r.	n.r.	n.r.
Sirisena et al., 2018 [43]	1	ped.	TDT	DFX	100%	0	100%	0	n.r.	n.r.
Alzaree et al., 2019 [28]	48	ped.	TDT	DFO	23/29%§	23/29%§	0	0	PTA, DPOAE	>20 dB in any frequency
Khan et al., 2019 [55]	198	mixed	TDT	DFX	45.5%	45.5%	n.r.	n.r.	PTA	n.r.
Khalaf et al., 2020 [56]	100	mixed	TDT	nr	8%	8%	n.r.	n.r.	n.r.	n.r.
Manara et al., 2021 [32] *	51	adult	TDT	mixed	58.8%	46%	5.9%	7.8%	PTA	>25 dB in any frequency
Manara et al., 2021 [32] *	20	adult	NTDT	mixed, none	35%	25%	5%	5%	PTA	>25 dB in any frequency

Abbreviations: TDT: Transfusion dependent Thalassemia; NTDT: Non Transfusion dependent Thalassemia; HL: hearing loss; SNHL: Sensorineural Hearing Loss; CHL: Conductive Hearing Loss; MHL: Mixed hearing Loss; PTA: pure-tone audiometry; PTAv: pure-tone average; ICT: iron chelation therapy; DFO: deferoxamine; DFX: deferasirox; DFP: deferiprone; Ca-DTPA: calcium-diethylene triamine penta-acetic acid; BAEP: Brainstem Auditory Evoked Potentials; DPOAE: Distortion Product Otoacustic Emissions; OAE: Otoacustic Emissions. * Disaggregate data according to beta-thalassemia phenotype are presented for Porter, Onerci, and Manara. ^ one patient on deferoxamine with conductive hearing loss due to otosclerosis is mentioned, without reporting whether TDT or NTDT. This patient was not counted in the prevalence rate. § data refer to right and left ear; data on patients were not available.

**Table 2 jcm-11-00102-t002:** Longitudinal studies and outcome at follow up.

Author and Year	#	Age	Phenotype	ICT	Follow-Up Length	Intervention	Outcome
Wonke et al., 1989 [34]	50	mixed	TDT	DFO	19 months §	change to Ca-DTPA	improved
Porter et al., 1989 [4]	47	mixed	mixed	DFO	2 years °	temporary withdrawal or reduction	improved or stable
Triantafyllou et al., 1990 [48]	120	mixed	TDT	DFO	n.r.	dose withdrawal or increased	stable
Cuda et al., 1991 [35]	50	mixed	TDT	DFO	3 years	none	worsened or stable
Sacco et al., 1994 [36]	36	mixed	TDT	DFO	5 years	dose reduction	recovered, worsened or stable
Kontzoglou et al., 1996 [23]	88	mixed	TDT	DFO	6 years	dose withdrawal or reduction	recovered, worsened or stable
Ambrosetti et al., 2000 [29]	57	adult	TDT	DFO	3 years	dose adjustment according to IOL	stable
Chen et al., 2005 [30]	25	mixed	TDT	DFO	2 years *	none	improved or stable
Shamsian et al., 2008 [17]	67	mixed	TDT	DFO	6 months	none or dose reduction	stable
Delehaye et al., 2008 [25]	60	mixed	TDT	DFO	20 months	n.r.	worsened or stable
Bhardwaj et al., 2016 [33]	30	n.r.	TDT	mixed ^	12 months	n.r.	worsened

TDT: transfusion-dependent β-thalassemia; ICT: iron chelation therapy; DFO: deferoxamine; IOL: iron overload; Ca-DTPA: calcium-diethylene triamine penta-acetic acid. § only in severe HL, * only for HL, ° only in SNHL, ^ deferoxamine, deferasirox or deferoxamine and deferasirox.

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
