# Peer review of "Hearing Loss in Beta-Thalassemia: Systematic Review"

_jcm, 2021, doi:10.3390/jcm11010102_

Round 1

Reviewer 1 Report

The authors present an interesting results of a systematic review of all the published studies on hearing assessment in β Thalassemia (Transfusion Dependent and Non transfusion Dependent) and included children and adults.

There are some points that the authors mention and I feel that I need to emphasize:

Control group: The lack of a control group.

Baseline studies: No studies analyzed the prevalence rate among patients naive for iron chelation and no baseline audiometry studies in newly diagnosed patients or at last before starting chelation as the authors mention "Since the outset of the hearing loss is still unpredictable, audiological testing is suggested before the start of treatment to assess the hearing threshold".

No study compared the rate of hearing loss among the three different chelators, there are few data on patients treated by Deferasirox (DFX) or Deferiprone as monotherapy.

The definition of hearing loss relate to different db thresholds, some of them quite low, below 25 db.

DFX previous treatment: it is not clear if patients treated with Deferasirox or Deferiprone were previously treated with DFO.

SNHL could also be a complication of beta thalassemia itself since at last one study found SNHL in naïve patients to chelators. Then baseline and regular audiometry follow up is required.

Author Response

Dear reviewer,

we would like to thank you for your positive comments.

We are happy that you emphasized the main critical topics, that need to be addressed in future studies.  Indeed these are the questions that we face in the daily clinical management and led us to carry out this review, to finally find out that they are still mainly unsolved. 

Best wishes,

Imma Tartaglione and Renzo Manara

Reviewer 2 Report

In the manuscript entitled "Hearing loss in beta-thalassemia: systematic review" Tartaglione et al. aim to address the question regarding hearing loss evidenced in the beta-thalassemia context. For this purpose, the authors have extensively and thoroughly gathered bibliography dated within a range of almost half a century. The manuscript is clear, well written and gives insight into this phenomenon, while pointing out the difficulty of such research. In addition, all limitations of this systematic review, along with the limitations of the initial studies have been properly described. This Reviewer only has some minor comments, as follow:

  1. Could you please rephrase the following sentence (lines 153-154): “However, some studies termed thalassemia intermedia also some patients under regular transfusion regimen (e.g. Chen et al. [35]) and the two classifications might present some minor discrepancies”.? It seems a bit difficult to follow.
  2. The authors state that in most studies there were no controls used. Since some studies report results in both TDT and NTDT subjects, could the authors please indicate which group they would consider a proper control for studying hearing loss in these patients?

Author Response

Dear reviewer,

thanks for your positive comments.

We ameliorated the present manuscript according to your thoughtful suggestions.

In particular, 1) we rephrased the highlighted sentence on phenotype classification (see Reviewer#2 comment and the revised manuscript) and 2) we specified “healthy” controls when appropriate in order to make clear that the missing data in the literature mostly refer to the general population.

In fact, the impact of beta-thalassemia (including its treatment) on hearing loss and tinnitus can be correctly driven only if the study also provides the regional prevalence in the non-thalassemic population, as hearing loss (either conductive, sensorineural or mixed) and tinnitus are not rare in the general population.

This is particularly true when dealing with conductive hearing loss, that might be transient in the pediatric population, or with tinnitus that is also not rare in the young adult population.

Best wishes,

Imma Tartaglione and Renzo Manara